# Candidate Genes for Salt Tolerance in Forage Sorghum under Saline Conditions from Germination to Harvest Maturity

**DOI:** 10.3390/genes14020293

**Published:** 2023-01-22

**Authors:** Shugao Fan, Jianmin Chen, Rongzhen Yang

**Affiliations:** School of Resources and Environmental Engineering, Ludong University, Yantai 264025, China

**Keywords:** sorghum, forage, genes, phenotype, salinity

## Abstract

To address the plant adaptability of sorghum (*Sorghum bicolor*) in salinity, the research focus should shift from only selecting tolerant varieties to understanding the precise whole-plant genetic coping mechanisms with long-term influence on various phenotypes of interest to expanding salinity, improving water use, and ensuring nutrient use efficiency. In this review, we discovered that multiple genes may play pleiotropic regulatory roles in sorghum germination, growth, and development, salt stress response, forage value, and the web of signaling networks. The conserved domain and gene family analysis reveals a remarkable functional overlap among members of the bHLH (basic helix loop helix), WRKY (WRKY DNA-binding domain), and NAC (NAM, ATAF1/2, and CUC2) superfamilies. Shoot water and carbon partitioning, for example, are dominated by genes from the aquaporins and SWEET families, respectively. The gibberellin (GA) family of genes is prevalent during pre-saline exposure seed dormancy breaking and early embryo development at post-saline exposure. To improve the precision of the conventional method of determining silage harvest maturity time, we propose three phenotypes and their underlying genetic mechanisms: (i) the precise timing of transcriptional repression of cytokinin biosynthesis (IPT) and stay green (stg1 and stg2) genes; (ii) the transcriptional upregulation of the SbY1 gene and (iii) the transcriptional upregulation of the HSP90-6 gene responsible for grain filling with nutritive biochemicals. This work presents a potential resource for sorghum salt tolerance and genetic studies for forage and breeding.

## 1. Introduction

The fact that global salinity is not only vast but expanding has progressed from a claim to reality. For example, when we run a confusion matrix of the previous five remote sensing studies that predicted global salinity events [1,2,3,4,5], three outcomes are clear: non-saline land will remain or become slightly saline, slightly saline land will remain or become moderately saline, and moderately, highly, and extremely high saline areas will remain so for an unpredictable future (Table 1). It is worth noting that the study results encompass marginal and uncultivated land on a greater scale. With increased improper irrigation techniques in agricultural land, it is reasonable to deduce that the pace of salinization in cultivated areas is higher. Sorghum is an important food, feed, and industrial crop with the potential to mitigate food insecurity in China and other parts of the globe. Because sorghum is a moderately salt-resistant crop, it is confronted with all three possibilities, and in order to survive, it must quickly adapt to the rapidly expanding salinity—which is doubtful.

Currently, breeders are combining conventional and modern phenomics, genomics, and environics to identify and select superior varieties in terms of yield and salt tolerance, as well as to generate high-yielding salt tolerant genotypes [6]. The limitation of the former is that it is time-consuming and necessitates the scanning of huge populations, and it is difficult to get accessions with all the desired features of interest. The latter results in fewer gene pools, which might result in a popular sire effect. The ongoing extension and rise of salinity, and how sorghum can deal with it in a systematic manner remains a bottleneck to tackle in breeding and modeling programs. The first step in addressing this is to deconstruct the whole-plant genetic process governing the phenotypic of interest, which includes salinity adaptation.

Damage to plants by salt stress involves ionic toxicity, osmotic imbalance, and oxidative damage. Except for Na and Cl toxicity, plants’ responses to salinity and drought at the genetic level are similar. The osmotic effect, for example, which is the initial phase of salinity stress, imposes physiological drought, eliciting a response comparable to drought stress. As a result, studies in sorghum involving osmotic stress caused by drought will also be reviewed here.

## 2. Seed Dormancy Release and Germination

The readiness of a seed to germinate for subsequent multiplication is essential for an efficient silage production program. As a result, understanding the dormancy breakage of sorghum seeds is required for timely germination. After seed dormancy, the germination and emergence phases of the sorghum life cycle are the most sensitive to salt stress; seedling characteristics are suggested as a viable criterion for genotype selection with better performance under saline conditions stress during the germination and seedling growth stages, which affect ultimate plant performance, will be critical for producing salt-tolerant and high-yielding genotypes [7].

Sorghum seed recovery from dormancy is influenced by abscisic acid (ABA) signaling and gibberellic acid (GA) metabolism [8]. Embryo susceptibility to ABA and GA was associated with dormancy expression [9]. For instance, sorghum expressing four positive regulators of ABA signaling, i.e., *SbABI3/VP1*, *SbABI4*, *SbABI5*, and *SbPKABA1*, exhibited an increase in the germination rate [10]. Greater embryo sensitivity to ABA and higher expression of *SbABI4* and *SbABI5* in dormant grains resulted in increased GA catabolism, which was related to early sorghum emergence from dormancy. Both *SbABI4* and *SbABI5* ABA signaling components interact with the *SbGA2ox3* promoter, which includes an ABA-responsive complex (ABRC) [11]. This suggests that the two genes may compete for the same cis-acting regulatory elements related to sorghum grain release from dormancy. Therefore, we dig deeper into the potential mechanism behind this transcriptional regulation. A comprehensive transcriptome analysis of several sorghum genes encoding putative GA enzymes, including *SbGA2ox3*, revealed a transient increase in these transcripts in more dormant sorghum grains as compared to less dormant grains [12]. Recently, it has been shown that repressing SbGA2ox3 resulted in active GA buildup, promoting dormant grain germination [13] and thus suggesting that *SbABI4-* and *SbABI5*-mediated transcriptional suppression of *SbGA2ox3* could be the mechanism underlying GA buildup and release from dormancy.

After dormancy release, similar to other plants, sorghum’s response to salt stress during the germination and seedling phases is a complex quantitative trait governed by pleiotropic genes and maintaining a high germination rate enhances its chances of survival under stress. Dubbed the ‘jack of all trades’, the *WRKY* genes are one of the most extensive groups of transcriptional regulators in plants which play critical roles in early plant development and salt stress response [14,15]. A cis-acting element analysis reveals that all sorghum *SbWRKYs* include at least one salt stress response-related cis-element, as well as early development phenotypes. Also, an in-depth expression analysis of individual genes shows a high upregulation of *SbWRKY50* and *SbWRKY56* during the early sorghum germination stages under salt stress [16]. The *SbWRKY50* binds to the upstream promoter of the HKT1 gene during the three-leaf phase of a sorghum seedling [17]. Sorghum *HKT1* gene functions to maintain optimal Na^+^/K^+^ balance under Na^+^ stress [18], implying that *SbWRKY50* has a role in sorghum salt response via regulating ion homeostasis at the germination stage. Another salt upregulated gene in the work of Baillo et al. is *SbWRKY56*. Interestingly, under salt stress, the overexpression of its ortholog *AtWRKY46* enhances salt-stress independent root growth and photosynthesis via the regulation of ABA signaling and auxin homeostasis in Arabidopsis seedlings [19]. This maintenance of photosynthesis at the seedling stage ensures steady source-sink carbon flow which is vital for plant survival and the NADP-malate dehydrogenase (NADP-ME) enzyme is essential for the C4 photosynthetic pathway. During the seedling stage, a sorghum *SbNADP-ME* overexpression increases salt tolerance and alleviates PSII and PSI photoinhibition by improving photosynthetic capacity [20]. These observations suggest that the co-expression of *SbWRKY50*, *SbWRKY56*, and *SbNADP-ME* at the seedling stage may enable sorghum to achieve maximum photosynthetic capacity while maintaining high ion homeostasis and root growth. Ion homeostasis and root growth were determinant factors in tomato seedling survival under salt stress [21]. Figure 1 shows transcriptional regulation of sorghum seed pre- and post-salt exposure.

## 3. Early to Late Salt Stress Signaling

### 3.1. Hormonal Signaling

Plant hormones play important roles in salt stress signaling and transduction [22]. One of the most effective response mechanisms to salt stress is the biosynthesis and accumulation of ABA, which is a key regulator in the activation of plant cellular adaptation to salinity [23]. The ABA biosynthesis pathway involves the action of the primary components of the ABA pathway, including gene members from the PYL/PYRs, PP2Cs, and NCEDs families [24]. Individual members of this family, i.e., *SbNCED5*, *SbPYL7*, *SbPP2C53*, *SbPP2C09*, *SbPP2C23* and *SbPP2C58* were overexpressed under salt stress in sorghum which correlated positively with ABA accumulation [25].

Our in-depth analysis reveals that during salt signaling, genes in the ABA pathway may interact directly with other signaling molecules or indirectly through transcriptional regulation to trigger a salt response. For instance, typically ABA has an abscisic acid (ABA)-responsive element (ABRE)-binding factors that regulate the expression of target genes involved in signaling to high salinity by binding ABRE cis-elements in the promoter regions [26]. An overexpression of ABREs in sorghum reveals a potential role in sugar signaling motifs under osmotic stress [27]. This implies that ABA signaling may participate in salt-triggered physiological drought-induced remobilization of sugar molecules in sorghum. It is interesting to realize that the NAC salt-responsive superfamily may also participate in ABA cascade salt signaling in sorghum. First, *SbNAC17*, *SbNAC46*, *SbNAC26*, *SbNAC56* and *SbNAC73* genes showed time-differential upregulation patterns in sorghum response to salinity [28]. During the germination stage, *SbNAC56* overexpression in transgenic lines exhibited significantly enhanced hypersensitivities to NaCl in an ABA-dependent manner [28]. An ortholog of *SbNAC26* is a rice *OsNAC3* which positively regulates salt tolerance and plant height through the ABA signaling cascade [29]. *SbNAC73*, *SbNAC46*, and *SbNAC17* belong to the NAP subfamily of the NAC superfamily. These genes are orthologous to Arabidopsis *AtNAP*, which is a conserved senescence promoter and functions as a regulator of salt-induced osmotic stress responses, through an ABA pathway-dependent gene AREB1 [30]. *SbNAC58* belongs to the ATAF subfamily of the NAC superfamily whose overexpression enhanced the shoot growth, RWC, and ABA-mediated salt tolerance [31]. This may infer that NAP and ATAF subfamilies of NAC play essential roles in sorghum response to salt stresses in the ABA-dependent signaling pathway and may contribute to the shoot dry mass.

Auxins are vital for shoot growth and plants’ response to salinity. When we scan through the cis-acting regulatory elements within promoter regions of sorghum auxin genes, we observe that all the gene families involved in auxin biosynthesis, i.e., *SbGH3*, *SbLBD*, *SbARF* and *SbIAA* are responsive to and upregulated by salt stress [32]. Further, an expression analysis reveals that highly salt-responsive *SbARF16* and *SbARF7* are also highly expressed during early, mid, and late physiological maturity in sorghum. Evolutionarily, these genes belong to the Class III clade and their orthologs are *AtARF19* and *AtARF12*, respectively [33]. *AtARF19* promotes flowering, stamen development, floral organ abscission and fruit dehiscence [34]. *OsARF12*, an ortholog of *SbARF7* which is abundant in the sorghum root, is implicated in regulating auxin-mediated root elongation and high iron content [35]. These observations shed light on ARFs genes that may not only be involved in auxin biosynthesis and signaling under salt stress but also influence nutrient acquisition via root elongation and enhance panicle development.

Ethylene is an important hormone that determines many aspects of the plant’s vegetative development. The Ethylene Response Factors (ERF) act as critical downstream components of the ethylene signaling pathway [36]. From a phylogenetic analysis, predicted orthologous genes for sorghum ERFs includes *OsDREB1A* (*SbERF027*), *OsDREB1B* (*SbERF025*), *OsDREB1C* (*SbERF100*), *SbERF085* (*OsDREB1D*), *OsDREB1E* (*SbERF072*), and *OsDREB1F* (*SbERF042*) whose promoter’s regions contained cis-elements related to salt stress response and response to ethylene [37]. It is not surprising that all the orthologous members of sorghum ERFs above have Dehydration-Responsive Binding Elements (DREB) domains. It is well established that these DREB genes have a conserved function in plants’ response to drought and salinity-triggered dehydration [38]. Our further functional analysis of sorghum genes with DREB domain reveals that root abundant *SbDREB2* under salt stress is targeted by putatively five miRNAs indicating their roles in post-transcriptional regulation [39]. *SbDREB2A* ortholog is a rice *OsDREB2B* [40] which regulates salt tolerance through the ABA-mediated pathways and enhanced shoot growth performance [41]. Further functional analysis reveals that a sorghum *SbERF094* ortholog regulates ethylene-mediated root to shoot signaling during salinity stress and promotes shoot growth in rice [42]. The induction of these genes by osmotic stress in sorghum indicates their potential involvement in salt-induced ethylene signaling and physiological drought stress response in sorghum.

The brassinosteroids (BR) are key regulators of plant development and physiology whose biosynthesis and signaling are mediated by the BES1 gene family. In sorghum, two BES1 genes (*SbBES1-4* and *SbBES1-9*) were abundant in the root and were upregulated under osmotic stress [43]. The BES1 genes are conserved in function and work synergistically to positively regulate BR signaling [44] and recent salt response [45]. Due to the scarcity of the literature on this gene in sorghum, we explore its possible interaction with other known genes. We observed that most genes in the BR signaling pathway in sorghum have a conserved bHLH domain. For instance, the salt responsive *SbBHLH050* in sorghum belongs to the BEE3 subfamily of the bHLH superfamily (others being BEE1 and BEE2), which is an early responder to BR signaling components. Friedrichsen et al. [46] and Moreno et al. [47] have shown that BEEs are strongly induced by salt stress and overexpressing plants exhibited larger leaf diameters, seed yield, stem length, and number of internodes. Another salt-responsive gene in sorghum is *SbBHLH079*, whose homolog in rice induced a wide leaf angle phenotype and produced long grains with enhanced BR signaling [48]. Taking these findings together, we infer that salt signaling in sorghum via the BR pathway includes the activation of BEE genes, the members of which have conserved bHLH domains and promote foliar development.

Other than seed dormancy breaking, the important role of GA in the response to abiotic stress is becoming increasingly evident. For example, reduction of GA levels and signaling has been shown to contribute to plant growth restriction on exposure to salt stress [49]. The complex pathways of bioactive GA biosynthesis in higher plants require three different classes of enzymes encoded by the GA gene family, including GA20 oxidase (GA20ox), GA3 oxidase (GA3ox), and GA2 oxidase (GA2ox) [50]. Under salt stress, *SbGA2ox1* was expressed throughout the sorghum life circle with a bias in the root compared to the leaves, indicating a possible root-soil interaction. *SbGA2ox2* exhibited the highest level at all developmental stages and in all tested tissues, indicating an interaction between the root and shoot. Interestingly, the expression pattern of *SbGA20ox3* had a higher expression level during early sorghum development. Biosynthesis of these GA was associated with stem biomass increase [51]. To further understand the potential role of GA signaling in salt response in sorghum, it is important to look at the transcriptional regulation of GA biosynthesis and accumulation in sorghum which eventually influences signaling. DELLA genes from the GRAS family are suggested to be the main transcriptional regulators of GA biosynthesis and signaling [52]. Three DELLA genes i.e., *SbGRAS68*, *SbGRAS03*, and *SbGRAS23* have been mapped in the sorghum genome [53]. Repression of these genes 9 days of sorghum growth shows high GA accumulation and improved germination rate [53]. The three genes are orthologs of the rice *OsGRAS1* gene, which has a promoter region containing salt resistance-related and hormone response cis-elements and has been shown to improve germination under salt stress [54]. This finding implies that GA-mediated salt signaling and emergence in sorghum occurs via transcriptional repression of three DELLAs, *SbGRAS68*, *SbGRAS03*, and *SbGRAS23*, which is consistent with prior findings that DELLA and GA operate in opposing ways [55].

### 3.2. Non-Hormonal Signaling

Ca2+ signaling and its downstream calcium-dependent protein kinases (CPKs) play critical roles in the detection and transmission of stress signals [56]. Studies on CPKs in sorghum and their roles in response to salt stress, on the other hand, are largely unexplored. *SbCDPK6*, *SbCDPK59*, *SbCDPK30* and *SbCDPK27* are highly expressed in sorghum under osmotic stress [57]. The mitochondrial calcium uniporter (MCU) is a Ca2+channel complex component that regulates intracellular Ca2+ signal transduction. Four *SbMCU* genes have been identified in the sorghum genome, and expression patterns analysis revealed that the genes were differentially expressed in different tissues, with *SbMCU5.2* exhibiting the greatest expression under osmotic stress [58,59]. Members of the MCU5.2 class have an impact on the downstream signaling pathways caused by osmotic stress, which include the activation of mitogen-activated protein kinases (MAPK) [60]. So far, 12 MAPKs have been discovered in the sorghum genome. All of the *SbMAPKs* were upregulated during early salt response except for *SbMAPK13*, whose expression peaked 12 h later [61]. We suggest that *SbMAPK13* could be involved in post-salt signaling events rather than signal sensing. Our phylogenetic analysis reveals that SbMPK13 is an ortholog of rice *OsMAPK2*, which is responsible for ABA-mediated salt response and was associated with enhanced phosphorus uptake and accumulation of the shoot [62]. This finding implies that salt-induced Ca2+ dynamics through the mitochondrial MCU may trigger a complex signaling cascade involving hormones and *SbMAPKs*. A coordination of these hormones from early to late plant development under salt stress have been suggested to influence the Salt Overly Sensitive pathway which influenced the circadian clock and production of floral phenotypes [63]. Figure 2 demonstrates the various signaling pathways in sorghum under salt stress.

## 4. Root Developmental Plasticity

The root capacity to support plant growth and development amid changing soil conditions can be critical in mitigating the effects of salt stress and maintaining higher crop productivity. Roots, on the other hand, are frequently disregarded in crop production strategies. Understanding the genetic aspects of root growth and developmental flexibility might provide an opportunity for sorghum breeders to generate varieties with more resistant system designs to salt stress.

A plant’s capacity to regenerate new root hairs is vital for mineral and water absorption, which are significant aspects of plant development and tolerance to salt stress. A sorghum *SbbHLH85* controls resistance to salt stress by enhancing root hair development [64]. The TCP is another prominent gene family with a conserved bHLH domain that plays robust roles in altering root growth to influence agronomic properties in crops. In sorghum *SbTCP10*, *SbTCP13*, and *SbTCP15* are among the genes that are highly expressed in the sorghum root at the seedling stage. An analysis of abiotic stress response reveals that the three genes are upregulated by Na+ [65]. The three genes are orthologous to an Arabidopsis *AtTCP14* whose expression enhanced radicle growth and was implicated in early root growth [66]. Reactive oxygen species (ROS) scavenging is essential for the maintenance of plant growth under salt stress. Salt stress greatly enhanced a sorghum *SbNAC2* whose heterologous overexpression enhanced antioxidant enzyme-mediated primary root growth [67]. This suggests the important role of the gene is mediating growth and ROS scavenging in sorghum roots.

Root apoplastic barriers, consisting of Casparian bands and suberin lamellae, play pivotal roles in blocking the apoplastic bypass flow of water and ions into the stele and Na+ transport into shoots [68]. Salt stress induces the strengthening of the root apoplastic barriers [69]. Twenty-four genes encoding enzymes involved in the Casparian strip production were expressed differentially in the sorghum root [70]. *SORBI3006G148800*, which encodes nutritive phenylalanine proteins that are also precursors to lignin, were among the genes that were upregulated in sorghum early root development. *SORBI3002G250000*, which participates in the synthesis of S-type lignin, was also upregulated throughout the sorghum root growth stages [71]. These genes belong to the CASP-like subfamily and are found on chromosomes 9 and 16, respectively, in the sorghum genome, and are thought to play a conserved role in water retention. Salt stress, for example, increases the expression of the sorghum *SbCASP4* gene, which is in the endodermis cells of sorghum roots. The heterologous expression of *SbCASP4* in transgenic Arabidopsis was related to increased water retention and resistance to salt stress Na repulsion [71]. Since aboveground lignin accumulation lowers the forage value of sorghum, it is interesting to note that here upregulating genes coding key root lignin precursors may trigger events that promote forage value aboveground through water retention.

Suberin biosynthesis and accumulation may serve as an adaptive response to prevent excessive water loss caused by salt-induced negative osmotic pressure. The Cytochrome P450 (CYP) gene family’s *CYP89A2* and *CYP89A*, which encode enzymes that play conserved roles in plant cell apoptosis, are among the most upregulated genes during early, mid, and late sorghum root development [72]. In addition, upregulated *SbGPAT5* is also thought to have a role in sorghum root growth. The gene codes for the enzyme GPAT5 catalyzes the transfer of an acyl group from an acyl donor during suberin biosynthesis [73]. Furthermore, *SbHHT1*, a significantly upregulated acyltransferase, is necessary for the incorporation of ferulate into suberin during root development [74]. These findings indicate multi-family involvement in suberin biosynthesis during different developmental stages of the sorghum root, functioning as an important barrier for water loss caused by salt-induced negative osmotic pressure contributing to high water content. In Arabidopsis the suberin content and protein structures discussed here constituted important aspects of suberin’s barrier function in lowering water loss and sodium absorption through roots for improved drought and salt stress performance [75]. Figure 3 summarizes our hypothetical apoplastic pathway in sorghum root under salt stress.

## 5. Water Absorption and Channeling

Plants can overcome salt-induced osmotic stress by accumulating suitable osmolytes. Sorghum overexpressing the proline biosynthesis genes *SbP5CSF129A* demonstrated better salt stress tolerance and high-water retention [76]. Furthermore, the capacity to manufacture and store glycine betaine is common in angiosperms and is hypothesized to contribute to salt stress tolerance. The overexpression of *SbBADH1* and *SbBADH15* genes, which encode Betaine Aldehyde Dehydrogenase proteins, greatly increased osmotic potential and allowed for maximum osmotic adjustment, enhancing water intake and aboveground performance under salt stress in sorghum [77]. Another essential suitable osmolyte is mannitol. The mannitol biosynthetic pathway was designed into a Sorghum cultivar SPV462 by introducing the mtlD gene, which encodes mannitol-1-phosphate dehydrogenase. Under salt stress, transgenic plants overexpressing the gene absorbed and retained more water and sustained greater shoot and root development when compared to untransformed controls [78]. These findings suggest that genes involved in the soluble sugar and protein production pathway may play a crucial osmoregulatory function in sorghum salt-induced osmotic stress by promoting osmotic adjustment and water absorption from the soil.

Following absorption, the water channeling network oversees delivering water throughout the plant in order to keep it hydrated. The aquaporin (AQP) gene family is among the most functionally conserved of all gene families for this role [79]. Palakolanu et al. [80] used a genome-wide approach to identify and characterize sorghum AQP genes. Results showed high salt-induced expression of aquaporins coding for tonoplast and plasma intrinsic proteins, i.e., SbTIP2;1, SbTIP3;1, SbPIP1;5, SbPIP1, PIP1;5, SbPIP2.8 and SbPIP1;2. Overexpression of *SbTIP2;1* homolog has been shown to increase growth and tolerance to salt stress by increasing water uptake and retention [81]. Also, the heterologous expression of a *SbPIP1;2* homologs in Arabidopsis increased water absorption and CO2 uptake under salt stress. Furthermore, rewatering following osmotic stress allowed a considerably larger percentage of transgenic plants to be recovered, showing the transgenic plants’ capacity to maintain water and viability under water stress [82]. PIP1;5 was shown to be important in enhancing root water absorption in sorghum during osmotic stress, while *SbPIP2.8* was linked to increased root water permeability [83]. Protoplasts from tobacco overexpressing a sorghum *SbPIP1* absorbed water faster than wild type retaining higher RWC [84]. These functional conserved roles of sorghum aquaporins indicate the overlapping role of tonoplast and plasma intrinsic to protein’s potential role in water channeling in sorghum to prevent salt-induced osmotic damage and promote photosynthesis. Overexpression aquaporins in a tomato revealed that they might help in recovery from salt injury both in the roots and leaves, implying their important roles in regulating water transport in an organ-specific manner [84].

## 6. Photosynthesis and Carbon Partitioning

In photosynthetic plants, the chlorophyll and carotenoid pigments constitute the light-harvesting complex (LHC) which is encoded by the nuclear LHC gene family. In sorghum, these LHCs are associated with antenna protein PsbR which plays key roles in the PSII core complexes [85]. PsbR-encoding genes *PsaK*, *PsaH*, and *PsaO* were upregulated by salt stress in salt-tolerant sorghum compared to sensitive ones and were associated with a greater light-harvesting capacity [86]. After light harvesting, the NADP-ME catalyzes the oxidative decarboxylation of malate to generate CO2, pyruvate, and NADPH, which is essential for the carbon fixation process in C4 plants [87]. *SbNADP-ME* gene expression in sorghum is highly enhanced by salt stress, and its overexpression in Arabidopsis exhibited improved photosynthetic parameters such as PSII photochemical efficiency [20], indicating that the gene may alleviate salt-induced damage in sorghum by enhancing photosynthesis.

After C fixation, C partitioning during photosynthesis is an important aspect in determining silage nutritional value, and sugar is the most common kind of carbohydrate involved in plant carbon transfer. The accumulation of sugars in sorghum necessitates the action of enzymes [88]. Sucrose synthase enzymes are known to have a role in sucrose synthesis in sorghum, with significant activity in source tissues such as stem internodes [89]. The enzymes are coded by SS family genes, whose members are more up-regulated in salt-tolerant sorghum during salt stress, resulting in greater sucrose accumulation in the stem [90]. These findings imply that the level of overexpression of sucrose synthase genes is critical in regulating sorghum’s tolerance to salt stress.

The remobilization of synthesized sugars from the source to sink tissues is an important process that determines forage yield in crops. Sugars will eventually be exported transporters (SWEET) is a newly identified family of sugar transporters that have been characterized in Arabidopsis and rice, while very little knowledge of sugar accumulation in sorghum is available [91]. Comprehensive transcriptome analysis of these genes has been done on sorghum by Mizuno et al. [92]. Most of the genes are related to sugar accumulation in sorghum stem compared to other plants and among the most upregulated is *SbSWEET8-1*. Phylogenetic analysis revealed that *SbSWEET8-1* is orthologous to Arabidopsis SWEET11 and SWEET12 [93]. The two genes are bidirectional sugar transporters that are involved in phloem loading by mediating export from parenchyma cells into the sieve companion cell complex, thus contributing to the sucrose migration from source to sink and osmotic balance [94]. *SbSWEET9-3* was highly expressed in the panicle and was grouped into the same clade as AtSWEET8/RPG1 which was reported to maintain the plasma membrane integrity of microspores [95]. It is well-known that plasma membrane integrity is an important aspect of salt tolerance. *SbSWEET2-1* and *SbSWEET7-1* were expressed and grouped in the same clade as rice *OsSWEET11/Xa13*, which is essential for early grain filling [96]. These observations indicate that SWEET transporters may play an important role by mediating source-sink dynamics but may also contribute to osmotic-adjustment mediated salt tolerance in sorghum. The important role of *SWEET* gene is sucrose accumulation and plant development which have been summarized by Mizuno et al. [92] and Guan et al. [95]. Figure 4 shows a hypothetical model of sugar partitioning in sorghum under salt stress.

## 7. Flowering and Pollination

Maintaining healthy flowers under salt stress is critical to sorghum’s reproductive performance and panicle yield. Nuclear factor Y (NF-Y) is an evolutionarily conserved trimeric transcription factor complex that consists of three subunits: NF-YA, NF-YB, and NF-YC. [97]. A total of 33 NF-Y genes have been identified in the sorghum genome and, under salt stress, all NF-Y subfamily members are upregulated in sorghum flowers [98]. These three subfamilies’ genes encode proteins that act as positive flowering regulators [99]. Most of the discovered *SbNF-Ys* in sorghum have orthologous relationships with rice and maize. For example, of *SbNF-YBs* and *SbNF-YB11* orthologs are OsHAP3H and OsHAP3C, respectively, which regulate photoperiodic flowering and grain development [100]. Interestingly, under osmotic stress conditions, transgenic maize overexpressing the ortholog of the *SbNF-YB11* gene shows tolerance to osmotic stress and maintenance of flower growth [101]. This suggests that this gene floral upregulation can promote sorghum flowering under salt-induced osmotic stress.

Genes of the trihelix family may also participate in flower development under salt stress in sorghum. *SbTH02* is classified into subfamily GT2 and has the highest expression levels in the sorghum pistils, which corresponds to its ortholog *AT5G03680.1*, which regulates collective leaf inflorescence development [102]. SbTH15 is also a member of subfamily GT2 and shares a motif composition with its Arabidopsis ortholog *AtTH26*. Previous studies have shown that *AtTH26* (*At5G28300*) is induced by NaCl and highly expressed in Arabidopsis inflorescence [103]. The highest floral expression in sorghum is observed in *SbTH07*, *SbTH10*, *SbTH14*, *SbTH25*, *SbTH33* and *SbTH36*. Further, a correlation expression analysis reveals a high correlation between the expression of *SbTH27* and *SbTH28*, *SbTH33* and *SbTH39*, *SbTH07* and *SbTH32*, and *SbTH07* and *SbTH10* in vital floral tissues such as stamen, pistil, and leaf pericarp of sorghum under salt stress [104]. The high floral specificity, significant correlation among their expression and osmotic stress responsiveness of these genes suggests their synergetic role in floral osmotic stress resistance.

During salt stress, the NAC family also appears to have a function in sorghum flowers. Phylogenetically, the *SNAC1* subfamily includes three sorghum genes (*SbNAC005*, *SbNAC021*, and *SbNAC052*) as well as several recognized NAC genes such as rice *OsSNAC1* (*Os03g60080*), maize *ZmSNAC1* (*JQ217429.1*), and wheat *TaNAC02* (*AY625683.1*) [105]. Osmotic stress raised the transcript levels of sorghum *SbNAC005*, *SbNAC021*, and *SbNAC052* in flowers [106]. Overexpression of *SNAC1* improved resistance to severe osmotic stress in transgenic rice reproductive tissues [31]. Thus, as we have suggested, the floral specificity of genes from nuclear factor Y, Trihelix, and NAC families under osmotic stress appear to be involved in flower water relations. To acquire a better understanding of this relatively new idea of floral hydration in sorghum, it is necessary to return to the extremely functionally conserved aquaporins. *SbPIP1;2* is likewise reported to be substantially upregulated in the sorghum flower under osmotic stress, and its ortholog was shown to play a vital function in water transfer from the papilla cell to the pollen during pollination, which increased floral hydration and overall RWC [107]. As a result, it is reasonable to conclude that sorghum flower hydration may be critical in overcoming salt-induced osmotic stress and sustaining reproductive success. It would be fascinating to learn how a *SbTH02* interacts with a *SbPIP1;2*, *SNAC1*, and a nuclear factor *SbNF-YB11* during flower development in sorghum under salt-induced osmotic stress. Figure 5 shows our hypothetical model for floral hydration under salt stress.

## 8. Silage Harvest Maturity

Predicting the duration between distinct sorghum development phases and the projected silage harvest maturity date is difficult. Sorghum should ideally be collected at the soft-dough stage. Conventionally, this is roughly determined at the stage between flowering and hard grain when the grain continues to grow; biochemicals are quickly accumulating in the kernel, the grain color transitions, older leaves continue to undergo senescence and lose the stay-green phenotype. The genetic basis of these events has received little attention, yet it is vital for modeling.

Stay green is an osmotic stress adaptation trait that is distinctively defined by a green leaf phenotype during grain filling or after sorghum flowering [108]. QTLs for osmotic stress have been found to coincide with loci for early leaf senescence, and there are several examples where increased osmotic stress tolerance was accomplished through selection for the stay-green phenotype in sorghum [109,110,111,112,113]. Cytokinin production boosts growth and productivity by increasing the foliar stay-green phenotype under osmotic stress which influenced grain filling and grain number [114]. This implies that the loss of the stay-green phenotype in sorghum can be tracked biochemically by measuring cytokinin levels and genetically by examining the expression of the cytokinin biosynthetic gene Isopentenyltransferase (IPT) and upregulation of *SbStg1* and *Sbstg2* genes. Transgenic tobacco overexpressing the IPT gene generated more trans-zeatin, delayed senescence, preserved more biomass, and revived following osmotic stress [115].

Grain color transition can also be used to determine the genetic basis of sorghum silage maturity. At the genetic level, the sorghum *SbY1* gene is the ortholog of maze Pericarp color 1 (*ZmP1*) gene, which is a transcriptional regulator involved in the flavonoid-related white-red grain color transition [116]. *SbY1* gene is putatively located on QTL qSTIP1, and there are polymorphisms of the Y1 locus between BTx623 and NOG backgrounds. Previously, the study reported loss of function in alleles BTx623 through a deletion, while NOG had no functional deletion [117]. This suggests that *SbY1* gene might be responsible for sorghum grain color transition changes.

Grain filling is another key biochemical factor that may influence sorghum silage harvest maturity. The relationship of grain starch accumulation with genomic areas encoding salt stress-related genes, membrane proteins, and putative signaling proteins reveals a more detailed participation of these sets of genes in sorghum grain filling mechanisms. For example, Sapkota et al. [118] discovered that the HSP90-6 gene interacts with many biosynthesis-related genes across the sorghum genome during grain filling, implying that this gene is likely a hub gene responsible for multiple pathways related to the processing and transportation of biochemicals during grain filling and warrants further research into its role in seed development at forage harvest maturity. Kamal et al. [119] observed a strong association between stay green genotype, grain filling stages, and panicle yield inn sorghum under salt stress. This suggests that simultaneous expression of genes in the stay green and grain filling pathway can play an important role in understanding the silage harvest readiness of sorghum.

As shown in Figure 6, we propose that biomarkers of three phenotypes at this stage should be designed and used to understand the genetic basis of harvest maturity. These are (i) the precise timing of transcriptional repression of cytokinin biosynthesis (IPT) and stay green (stg1 and stg2) genes which initiate early senescence events post flowering; (ii) the transcriptional upregulation of the *SbY1* gene which initiates grain color transition post-flowering; and (iii) the transcriptional upregulation of *HSP90-6* gene responsible for grain filling with nutritive biochemicals.

## 9. Conclusions and Future Perspective

These findings indicate that salt stress responses in sorghum involve complex chain molecular processes that include the interaction of regulatory proteins and the expression of target genes. Although research on the genetic mechanisms of ionic imbalances and oxidative components of salt stress is sufficient to draw conclusions, most studies have leaned toward the osmotic stress side of salinity. This is due, in part, to the fact that saline and aridity frequently coexist. More genes in the antioxidant and K/Na homeostasis pathways should be studied for a more comprehensive picture. Table 2 summarizes the gene families discussed here.

## Figures and Tables

**Figure 1 genes-14-00293-f001:**
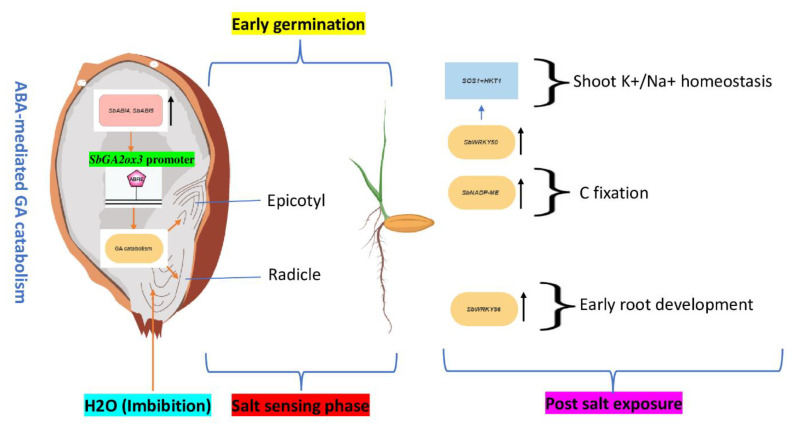
Schematic diagram illustrating transcriptional regulation of GA catabolism during sorghum seed dormancy release and post-salinity events. The black arrow pointing up is upregulation.

**Figure 2 genes-14-00293-f002:**
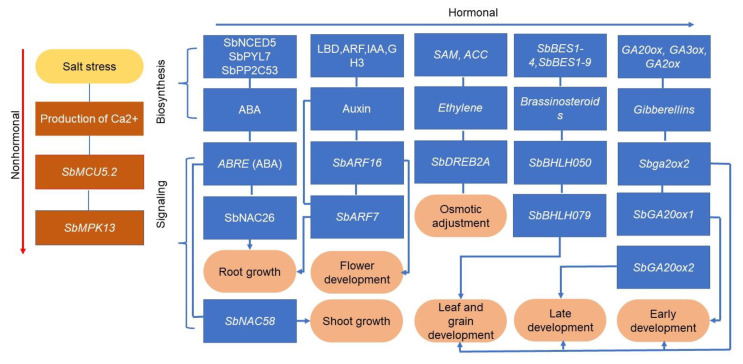
A flow diagram demonstrating the hormonal and nonhormonal salt signaling routes in sorghum, as well as the associated phenotypes.

**Figure 3 genes-14-00293-f003:**
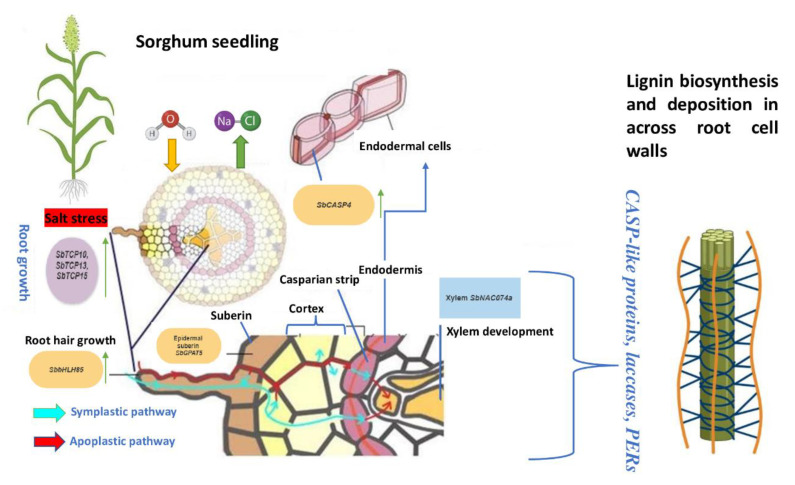
A schematic representation of the genetic regulation of the apoplastic and symplastic pathways in sorghum root under salt stress, as well as the corresponding phenotypes.

**Figure 4 genes-14-00293-f004:**
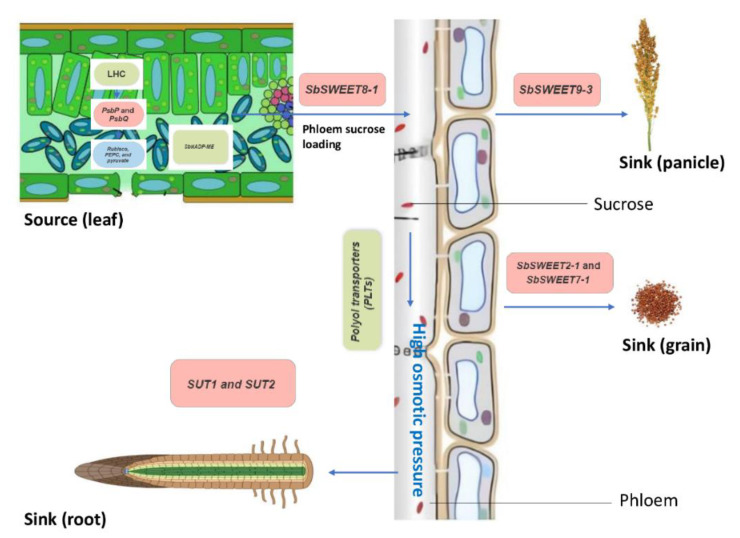
Our schematic depiction of source-sink carbon flux and their transporters in sorghum under salt stress.

**Figure 5 genes-14-00293-f005:**
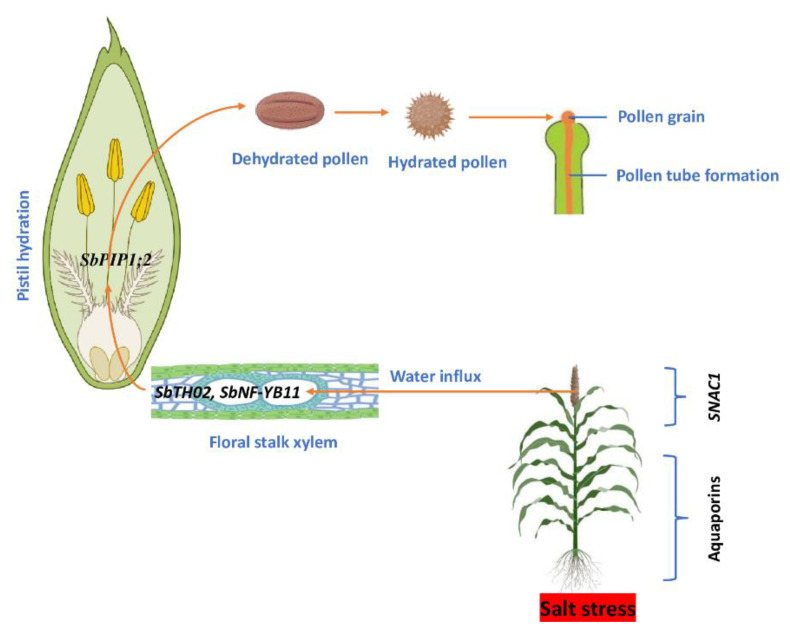
A schematic representation of floral hydration and its regulatory genes in sorghum under salt stress.

**Figure 6 genes-14-00293-f006:**
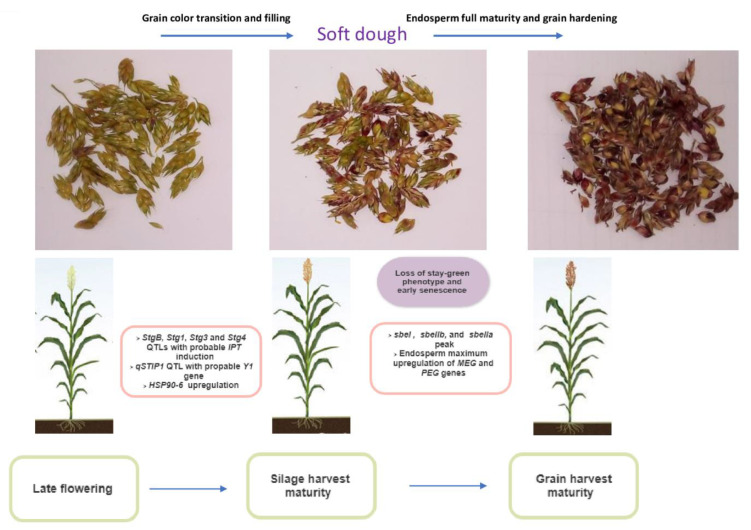
Figure depicting the phenotypic and underlying genetic process that may be utilized to simulate silage harvest maturity biomarkers to maximize production.

**Table 1 genes-14-00293-t001:** A computed confusion matrix and accuracy statistics of world salinity level.

Predicted							
Measured		**Salinity Level**	**Non**	**Moderately**	**Highly**	**Extremely**	**Total**
	Non	90	10	0	0	0	100
	Slightly	10	87	1	0	0	100
	Moderately	11	26	69	0	0	100
	Highly	15	34	3	47	0	100
	Extremely	18	29	1	0	49	100
	Total	144	186	74	47	49	500

*Note.* Measurements and predictions were conducted from five studies in 1986, 2002, 2005, 2009, and 2016.

**Table 2 genes-14-00293-t002:** Summary of gene families regulating forage phenotype in sorghum under salt stress.

Gene Family	Gene Name	Prediction Method	Growth Stage	Organ	Mode of Action	Reference	Phenotype
Gibberellins biosynthesis (GA)	*SbGA2ox3*	Transgenic	dormancy breaking and early germination	Seed	*SbABI4* and *SbABI5* mediated ABA signaling	Rodrguez et al. 2009	Promote early seed germination
WRKY	*SbWRKY50*	Transgenic	Seedling	Root	directly bind to the upstream promoters of SOS1 and HKT1.	Song et al. 2020	Promote K/Na homeostasis
	*SbWRKY56*	Orthology	Seedling	Root	Promotion of ABA-mediated auxin homeostasis	Ding et al. 2015	Root growth
bHLH	*SbbHLH050*	Transgenic	Seedling	Root	Salt induced induction of root hairs	Friedrichsen et al. 2002	Root hair growth
	*SbBHLH079*	Orthologs	Post flowering	Grain	early response BR signaling components	Seo, Hyoseob et al. 2020	Shapes the grain architecture
	*SbTCP10, SbTCP13, SbTCP15*	Orthologs	Throughout life cycle	Root	Radicle growth promotion	Tatematsu et al. 2008	Early sorghum root development
NAC	*SbNAC074a*	Orthologs	Seedling	Root	Differentiation of xylem tissue		Promotion of water transport
	*SbNAC56*	Transgenic	Seedling		ABA mediated hypersensitive to NaCl	Kadier et al. 2017	Root and shoot growth
	*SbNAC58*	Transgenic	Seedling		ABA mediated sensitivity to osmotic stress	Seok et al. 2017 Hu et al. 2006	Improved water intake
	*SbNAC005, SbNAC021 and SbNAC052*	Transgenic and orthology	Post flowering	Flower	Osmotic response	Sanjari et al. 2019, Hu et al. 2006	Improved water intake
Cytochrome	*SORBI3006G148800, SORBI3006G148900*	Orthologs	Emergence	Root	Conversion of phenylalanine to cinnamic acid and tyrosine to p-cinnamic acid	Yang et al. 2017	Casparian strip development
	*SbCASP4*	Transgenic	Germination	root	Involved in the phenylpropanoid pathway	Wei et al. 2021	Lignin biosynthesis
	*SbGPAT5*	Orthologs	Germination	Root	Catalyzes the transfer of an acyl group from an acyl donor to the sn-1 position of glycerol 3-phosphate	Murata et al. 1997	Suberin biosynthesis
ARF	*SbARF16, SbARF7*	Orthologs	Flowering	Flower	Floral organ abscission	Qi et al. 2012	Sorghum panicle development
ERF	*SbERF080, SbERF094*	Orthologs	Early to late		Ethylene mediated root to shoot salt signaling	Schmidt et al. 2014	Increased osmotic adjustment and water absorption
	*SbDREB2A*	Orthologs	Early to late	Leaves	ABA-mediated transcriptional regulation of drought responsive elements	Herath (2016)	Shoot growth
BES1	*SbBES1-4 and SbBES1-9*	Orthologs	Early to late	Roots	Work synergistically with BHLH family members under salt stress	Jia et al. 2021	Work synergistically to positively regulate BR signaling and salt stress tolerance
MCU	*SbMCU5.2*	Orthologs	Early to late	Root	Activation of mitogen-activated protein kinases (MAPK)	Teardo et al. 2019	Intracellular Ca2+ signal transduction and cationic homeostasis
MAPKs	*SbMAPK13*	Orthologs	Early to late	Root	Late ABA-mediated salt response	Yu et al. 2011	Stress-stimulus-specific Ca2+ dynamics in the chloroplast
Aquaporins	*SbTIP2;1*	Transgenics	Early to late	Shoot	Regulating the water and oxidative status	Martins et al. 2017	Increases in relative water content
	*SbPIP1;2*	Transgenic	Early to late	Leaves	Codes for plasma membrane intrinsic proteins		Increased root and leaf water
	*SbPIP2.8,*	Transgenic	Early to late	Root	Improves root permeability to water	Sun et al. 2017	Increasing the ability to retain water
Trihelix	*SbTH02*	Orthologs	Flowering	Flower	Stamen development	Shabalina et al. 2010, Frerichs et al. 2016	Leaf inflorescence development
	*SbTH15*	Orthologs	Flowering	Flower	Salt-induced floral differentiation	Xi et al. 2012	Flower development
Nuclear factor Y	*SbNF-YBs, SbNF-YB11*	Orthologs	Flowering		Regulates photoperiodic flowering	Wei et al. 2010	Flower development
SWEET	*SbSWEET8-1*	Orthologs	Soft dough	Shoot	Bidirectional sugar transporters	Eom et al. 2015, Chen et al. 2012	Phloem loading and sugar partitioning
	*SbSWEET9-3*	Orthologs	Flowering	Panicle	plasma membrane integrity	Guan et al. 2008	Source-sink (panicle) sugar transportation
	*SbSWEET2-1, SbSWEET7-1*	Orthologs	Soft dough	Grain	Sucrose release from maternal tissue to the maternal-filial interface	Ma et al. 2017	Source-sink (seed) sugar transportation

## Data Availability

Data sharing is not applicable to this article.

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
