# Peer review of "Candidate Genes for Salt Tolerance in Forage Sorghum under Saline Conditions from Germination to Harvest Maturity"

_genes, 2023, doi:10.3390/genes14020293_

Round 1
Reviewer 1 Report
good paper
Author Response
I sincerely thank you for taking the time to review our paper and your recognition of our research. We have carefully checked the spells and grammer errors.
Reviewer 2 Report
Comments and Suggestions for Authors
The authors investigate the "Candidate genes for salt tolerance in forage sorghum under saline conditions from germination to harvest maturity". The subject is interesting. However, the data seem only partly ok. In addition, the presentation and overall outlay of the manuscript require substantial revision.
1- Add the morpho-physiological discussions to each part, especially.
2- check and arrange the format of the reference based on the journal.
Author Response
point 1: The authors investigate the "Candidate genes for salt tolerance in forage sorghum under saline conditions from germination to harvest maturity". The subject is interesting. However, the data seem only partly ok. In addition, the presentation and overall outlay of the manuscript require substantial revision.
Add the morpho-physiological discussions to each part, especially.
Response: Thank you so much for the constructive suggestions. I really appreciate the helpful advice. The entire review primarily focuses on published sorghum genes putatively associated to phenotypes of forage value and salt performance. This includes but not limited to morpho-physiological characteristics. As a result, our critical discussions are guided entirely by the accessibility of these sorghum, genes in online reports to clearly emphasize the research gaps and propose a hypothesis. Though not in sorghum, we have however added some discussion from more reports that we think can provide further insights as per your suggestion. Thank you
2- check and arrange the format of the reference based on the journal.
Thank you so much. We have changed the entire references to suit the journal
Reviewer 3 Report
In the revies entitled "Candidate genes for salt tolerance in forage sorghum under sa-line conditions from germination to harvest maturity", the authors have done a good piece of study.
1- one minor revision is to add the line of future outcome in the abstract.
Author Response
Point1: In the revies entitled "Candidate genes for salt tolerance in forage sorghum under sa-line conditions from germination to harvest maturity", the authors have done a good piece of study.
one minor revision is to add the line of future outcome in the abstract.
Response: Thank you so much. We have addressed this by adding this statemen;
This work presents a potential resource for sorghum salt tolerance and forage genetic studies and breeding.
Round 2
Reviewer 2 Report
accept.